# Monitoring the trends of *Angiostrongylus cantonensis* infection in humans and *Pomacea* spp. Snails in Dali, Yunnan, China, 2007–2021

Tian-mei Li[1,2], Yu-hua Liu[1], Wen Fang[1], Shen-hua Zhao[1], Ting Li[1], Ling Jiang[1], Peter S. Andrus[2,3], Yun-hai Guo[2]*, Shao-rong Chen[1]*

**1** Institute of Schistosomiasis Prevention and Control, Dali Prefecture, Dali, Yunnan Province, China, **2** National Key Laboratory of Intelligent Tracking and Forecasting for Infectious Diseases, National Institute of Parasitic Diseases, Chinese Center for Disease Control and Prevention, Chinese Center for Tropical Diseases Research, Key Laboratory on Parasite and Vector Biology, Ministry of Health, WHO Centre for Tropical Diseases, National Center for International Research on Tropical Diseases, Ministry of Science and Technology, Shanghai, China, **3** School of Global Health, Chinese Center for Tropical Diseases Research, Shanghai Jiao Tong University School of Medicine, Shanghai, China

* guoyh@nipd.chinacdc.cn, dlxfcsr@163.com

## Abstract

We summarize historical events related to angiostrongyliasis and analyze surveys of clinically diagnosed and suspected cases of angiostrongyliasis in Dali from 2007 to 2021. We also randomly tested market sold *Pomacea* spp. snails to detect whether *Angiostrongylus cantonensis* was present in Dali market stalls from 2008 to 2021. There were a total of 125 cases of angiostrongyliasis (92 clinically diagnosed and 33 suspected) reported in the Dali Prefecture from 2007-2021. Of the 125 cases, 72 patients from 2010 to 2021 were investigated, with the main clinical manifestations being headache (100%), muscle pain (61%), neck stiffness (58.3%), paresthesia (58.3%), fever (55.5%), nausea (48.6%), coughing (26.3%), vomiting (44.4%), photophobia (18%), diplopia (25%), and visual impairment (5.5%). Laboratory testing showed cerebrospinal fluid qualitative protein levels and blood eosinophil levels were abnormal in 100% and 87.5% of patients tested, respectively. Moreover, of the 49,970 *Pomacea* spp. checked for *A. cantonensis* infection, 373 (0.75%) were found infected. Our study highlights the importance of enhancing public education, stricter food safety measures, and improved diagnostic methods to help mitigate future outbreaks of angiostrongyliasis.

## Author summary:

Human angiostrongyliasis is an emerging parasitic disease endemic in developing countries. It is commonly caused by the rat lungworm, *Angiostrongylus cantonensis*, which is currently a public health concern in Dali, Yunnan Province,

**Data availability statement:** All data is given within the manuscript itself. Sequence data is provided in GenBank accession numbers 'PP735489, OR816070-OR816087' for COI and 'OR790453-OR790469' for ITS2.

**Funding:** This work was supported by the Key Research and Development Program in Hainan Province (ZDYF2024SHFZ083 to YHG), the National key R & D project (2021YFC2300800 to YHG, 2021YFC2300804 to YHG), the Technology Innovation Support Program, NIPD, China CDC (TF2024012 to YHG), the Key Laboratory of parasitic Pathogens and Vector Biology of the National Health Commission (NHCKFKT2021-12 to TML), and the High-level health technical personnel in Yunnan Province (H-2018073 to TML). The funders had no role in study design, data collection and analysis, decision to publish, or preparation of the manuscript.

**Competing interests:** The authors have declared that no competing interests exist.

People's Republic of China, as well as other places in Asia and the Pacific. Our study examines the epidemiology and control measures from 2007 to 2021, during which 125 cases (92 diagnosed, 33 suspected) of human angiostrongyliasis were reported in Dali. Patients experienced many clinical symptoms, with the most common ones being headache, muscle pain, neck stiffness, paresthesia, fever, and nausea. Standardized laboratory tests revealed abnormal cerebrospinal fluid protein levels in all patients tested (n = 72), with the majority of patients showing elevated blood eosinophil levels. In addition to humans, *Pomacea* spp. snails sold in a local Dali market, were screened for *A. cantonensis* infection, with 373 of the 49,970 snails testing positive for infection. Analysis of the monthly averages of angiostrongyliasis cases and *Pomacea* spp. infection prevalence revealed a similar trend, with the highest incidence occurring during the spring months (February to May). Human angiostrongyliasis outbreaks in Dali has prompted the implementation of public health interventions, leading to enhanced monitoring and control efforts.

## Introduction

Human angiostrongyliasis is a zoonotic disease caused by the parasitic nematode genus, *Angiostrongylus* (Chromadorea: Angiostrongylidae). *Angiostrongylus* was first documented in rats in Guangzhou, China in 1935, and is commonly found in many tropical regions around the world [1]. *Angiostrongylus* nematodes use various gastropod species as their intermediate hosts, and rats as their definitive hosts [2]. Rats become infected by consuming infected gastropod flesh containing third-stage larvae (L3). The L3 larvae then migrate from the stomach to the brain and mature into adults, causing lesions. The adult worms then mate and release eggs into the bloodstream, which travel to the lungs and hatch into first-stage larvae (L1). The L1 larvae then migrate to the trachea and are excreted in the feces. Gastropods become infected by ingesting these L1 larvae and can transmit the parasite to humans, who serve as accidental hosts and cannot complete the nematode's life cycle [2].

The majority of human angiostrongyliasis cases are caused by *Angiostrongylus cantonensis*, which causes eosinophilic meningitis, ocular angiostrongyliasis, and rarely, encephalitic angiostrongyliasis [3]. Other species such as *A. costaricensis* cause abdominal angiostrongyliasis in South America, while the South Asian species *A. mackerrasae* and *A. malaysiensis* are also considered potentially pathogenic to humans [2]. Human cases of angiostrongyliasis occur when people unintentionally ingest a third stage larvae of *Angiostrongylus* by either consuming uncooked gastropods (or other paratenic hosts), eating raw contaminated vegetables, or drinking non-boiled contaminated water [4]. Unlike other gastropod-borne parasites, *Angiostrongylus* does not rely on one specific genus of gastropod as it can be transmitted through most freshwater and terrestrial gastropod species (intermediate host), as well as amphibians, chilopods, crustaceans, fish, planarians and reptiles (paratenic hosts) [5,6]. The current distribution of *A. cantonensis* and the number of angiostrongyliasis

cases reported are concentrated in tropical regions of South-east Asia (China, Malaysia, Thailand etc.) and numerous Pacific islands (Cook Islands, Fiji, Hawaii, New Caledonia, Vanuatu etc.) [2]. The first angiostrongyliasis outbreak recorded in mainland China, was in 1997 in Wenzhou, with 30 subsequent mass outbreaks occurring since then [7–9]. Through retrospective investigation, combined with clinical manifestations, epidemiology, etiology, and serum immunology, it has been confirmed that the cause of the outbreaks was food-borne, mainly caused by the ingestion of raw, or under cooked snails. The first national survey of *A. cantonensis* prevalence in Chinese gastropod species found two invasive species, *Pomacea canaliculata* (Gastropoda: Ampullariidae) and *Lissachatina fulica* (Gastropoda: Achatinidae) were commonly infected and widely distributed in the southern provinces of China [10]. These findings demonstrated that *P. canaliculata* and *L. fulica* snails are highly compatible with *A. cantonensis,* when compared to other freshwater and terrestrial gastropod species [10].

Before 2003, there were no reported cases of human angiostrongyliasis in the Yunnan Province. However in September 2003, 28 people in Kunming City experienced eosinophilic meningitis, which were characterized by headaches, fever, muscle pain and a stiff neck [11]. These 28 cases were ultimately diagnosed as angiostrongyliasis and were caused by the consumption of raw snail meat containing *A. cantonensis* larvae. As a result, human angiostrongyliasis was listed as an emerging food-borne parasitic disease by the World Health Organization (WHO), and was classified as an emerging infectious disease by the Chinese Ministry of Health in 2003 [12]. In November 2007, Dali City, Yunnan Province reported the first case of angiostrongyliasis, and subsequently detected the third stage larvae of *A. cantonensis* in commercially available *Pomacea* spp. snails sold in markets. As a response to these incidences, the government and party committee of the Dali prefecture implemented effective measures to control the epidemic of angiostrongyliasis. Additionally, these outbreaks of angiostrongyliasis in the Dali Prefecture, had a positive impact on the prevention and control of other gastropod-borne and food-borne parasitic diseases in the region (e.g., amphistomiasis, clonorchiasis, echinococcosis, trichinellosis etc.). Alongside the local Dali Prefecture government, the Chinese Center for Disease Control and Prevention (CDC) also classified the outbreak of angiostrongyliasis in the Dali Prefecture as a public health emergency. As a result, starting in March of 2008, the Dali region included angiostrongyliasis in the management and key monitoring of infectious diseases and categorized it as Class C [13].

In 2009, according to the national monitoring work plan (Notice of the Chinese Center for Disease Control and Prevention on Issuing the Pilot Work Plan for Symptom Monitoring and Transmission Early Warning of Angiostrongyliasis; No. 466), the Dali region has been listed as one of the three pilot testing areas for the monitoring and early warning of angiostrongyliasis transmission in the Yunnan Province [14]. The prevention and control of angiostrongyliasis has become a long-term task in Dali, Yunnan. In order to outline and understand the angiostrongyliasis epidemic in Dali, Yunnan, and to explore whether effective prevention and control strategies have been implemented, this study aims to outlines the current status and trends of angiostrongyliasis in Dali, Yunnan, from 2007 to 2021.

## Materials and Methods

### Ethics statement

Ethical approval (project number: 2021YFC2300800; approval number: 2021019) was given by the 'National Institute of Parasitic Diseases, Chinese Center for Disease Control and Prevention' ethical review committee. The study utilized anonymous clinical data, such as age, gender, occupation, and laboratory test results from human patients with suspected and confirmed *Angiostrongylus cantonensis* infection between 2007 and 2021. The anonymous clinical data used in this study were provided by the 'Dali Prefecture People's Hospital', the 'Dali First People's Hospital' and the 'Dali Prefecture Schistosomiasis Control Center'.

### Collection and organization of data sources

All patients included in this study sought medical care independently, presenting themselves to hospitals in the Dali Prefecture due to food-borne illness. The diagnoses of *Angiostrongylus cantonensis* infection were subsequently confirmed

by the respective hospitals. Human cases of angiostrongyliasis in the Dali Prefecture were recorded from 2007 to 2021, with a disease surveillance case study questionnaire given to 72 patients from 2010 to 2021. Excel was used to record and organize the number of reported human angiostrongyliasis cases in the Dali Prefecture, as well as to summarize and analyze the data.

## Diagnostic standards and laboratory testing methods for angiostrongyliasis cases

When human angiostrongyliasis cases were reported, several laboratory tests (blood count, cerebrospinal fluid test, chest X-Ray, Head CT scan and ELISA) were performed to diagnose *A. cantonensis* infection. The diagnostic methods used to confirm cases of human angiostrongyliasis were based on the 'Diagnostic criteria for *Angiostrongylus cantonensis*' diagnostic manual produced by The People's Republic of China Health Industry Standards (WS321–2010) [15]. This manual outlines the diagnostic tests required to confirm the clinical diagnosis of both suspected and confirmed cases of human angiostrongyliasis. It details the procedures for performing and interpreting complete blood count tests, immunological tests (e.g., ELISA), and the necessary steps to examine a patient's cerebrospinal fluid, eyes, brain tissue, and pulmonary arteries for the presence of *A. cantonensis* infection. All reported cases of angiostrongyliasis were diagnosed and verified at either the Dali Prefecture People's Hospital, the Dali First People's Hospital, or the Dali Prefecture Schistosomiasis Control Center.

### *Angiostrongylus* Enzyme-linked Immunosorbent Assay (ELISA)

The ELISA testing was conducted using a non-commercial, standardized '*Angiostrongylus* IgG Antibody Detection Kit' produced by Shenzhen Kangbaide Biotech Company (Shenzhen, People's Republic of China). Blood serum samples were prepared by diluting them with Reagent No. 5 at a ratio of 1:100 (e.g., 5 μL serum + 495 μL diluent), ensuring thorough mixing. Two controls were included in each run, a negative control (serum from an uninfected patient) and a positive control (serum from an infected patient). The controls were used undiluted and were added directly to the wells. For each test, 100μL of the diluted serum, negative control, and positive control were dispensed into their respective wells. The plate was incubated at 37°C for 30 minutes in the dark. Following incubation, the wells were washed three times with a washing solution, allowing the solution to sit for one minute during each wash. Subsequently, 50μL of Reagent No. 1 was added to all wells. The plate was again incubated at 37°C for 30 minutes in the dark. The washing procedure was repeated, after which 50μL of Reagent No. 3 and 50 μL of Reagent No. 4 were added to each well. The plate was gently mixed and incubated at 37°C for 10 minutes in the dark. The reaction was stopped by adding 50μL of stopping solution (Reagent No. 6) to each well.

The optical density (OD) of the wells was measured using a microplate reader set to 450nm, with 630nm as the reference wavelength. The plate reader was calibrated to zero using a blank control solution. The cut-off value (CoV) for a positive result was calculated as 2.1 standard deviations above the mean OD value of the negative control (mean + 2.1 S.D.), with a baseline value of 0.10 used if the negative control OD was ≤ 0.10. The test was considered invalid if the negative control OD exceeded 0.15, or if the positive control OD was less than 0.50. The time since symptom onset was not recorded for each patient. Serum from suspected patients was exacted and tested once upon hospital admission.

## Monitoring *Pomacea* spp. snails for infection

The prevalence of infected *Pomacea* spp. sold in Dali City wet markets was investigated monthly from 2008 to 2021. This was done by sampling *Pomacea* spp. snails from two fixed market stalls (1 kg from each stall) at a local market, twice a month throughout the year. If an infected snail was found, then the market stalls were sampled and monitored every week, until no infection was found. Infection was determined using the lung microscopy method [16,17]. In brief, this was done by removing the lung sac from each *Pomacea* spp. and examining the internal layers for *A. cantonensis* larvae.

## DNA extraction and PCR amplification

When infected *Pomacea* spp. were found, single *A. cantonensis* larva from a single *Pomacea* snail was isolated and their DNA was extracted using a Blood/Cell/Tissue Genomic DNA Extraction Kit (Tiangen, Shanghai, China). Once extracted, the Cytochrome C Oxidase Subunit I (COI) and Internal Transcribed Spacer 2 (ITS2) gene fragments were amplified using the Ac-COI (Forward: 5'-GGA TGT GGG ACT AGT TGG ACT G-3' and Reverse: 5'-TGA TGA GCC CAA ACC ACA CA-3') and Ac-ITS2 (F: 5'-ACG TCT GGT TCA GGG TTG TT-3' and R: 5'-TTA GTT TCT TTT CCT CCG CT-3') primer sets. All PCR reactions were performed using 12.5μl of 2x Taq Master Mix, 10.5μl of water, 0.5μl of forward and reverse primers and 1μl of DNA template. Two PCR controls were used for each run, a negative control (water) and a positive control (adult Angiostrongylus cantonensis DNA). The PCR cycling conditions used for both primer sets were as follows: pre-denaturation at 94°C for 3min, followed by 35 cycles of denaturation at 94°C for 1min, annealing at 50°C for 30secs, extension at 72°C for 1min, and a final extension stage at 72°C for 7min. For problematic samples, a low-stringency PCR was performed, wherein the annealing temperature was reduced to 48°C. All PCR products were electrophoresed on a 1% agarose gel, with successfully amplified PCR products being purified and sequenced in the forward direction using Sanger sequencing by Sangon Bioengineering (Shanghai).

## Phylogenetic analysis

The COI and ITS2 sequences had their forward and reverse primers removed, they were cut down to match the base pair length of their respective GenBank references and all sequences were aligned using MEGA 11, with misaligned sections of the COI and ITS2 being fixed manually. Phylogenetic trees were constructed using the maximum likelihood method, using a general time reversible model incorporating gamma rate correction (GTR+Γ) in the program MEGA 11, with bootstrap analysis undertaken using 1000 replicates.

## Results

### Regional, population and monthly distribution of angiostrongyliasis cases

In total, 125 cases of angiostrongyliasis were recorded in Dali from 2007 to 2021, with 46 cases being reported from 2007 to 2008 (Dali city: 36; Binchuan County: 3; Eryuan County: 2; Midu County: 1; Jianchuan County: 1; Weishan County: 1; and 2 cases outside the Dali Prefecture; Fig 1a). No cases were reported in 2009, but an additional 79 cases were reported in Dali city from 2010 to 2021 (Fig 1a). Among the 125 cases reported from 2007 to 2021, 92 were clinically diagnosed and 33 were suspected cases of angiostrongyliasis. Of the 125 patients reported, 61 were factory workers, 31 were farmers, 16 were students and 17 were office workers. Of the 92 clinically diagnosed cases, 49 of the patients were female and 43 were male. When we surveyed 72 of the 92 clinically diagnosed patients, we found 88.9% of patients (64/72) had a history of consuming raw or sauce fried snail dishes, with 66.7% (48/72) regularly eating these dishes at restaurants. The majority of the patients reported (84.1%) were between the ages of 20 and 60, with the youngest age being seven years old and the eldest age being 72 years old (0–5 years: 0 patients; 6–12 years: 7, 13–18 years: 5, 19–45 years: 90, 46–60 years: 17, and over 60 years: 6). The largest number of cases occurred in 2008, and again in 2011 due to an outbreak of angiostrongyliasis caused by *Pomacea* spp. infected with *A. cantonensis*. (Fig 1a). The incidence of angiostrongyliasis cases showed an obvious element of seasonality, with the number of cases concentrating in the spring (February to April), with the highest incidence occurring in February (Fig 1b).

### Main clinical manifestations experienced by patients

The first outbreak of angiostrongyliasis occurred in the Dali area from late 2007 to early 2008. As a response to this outbreak, improvements were made to the health system and 72 diagnosed cases of angiostrongyliasis from 2010 to 2021

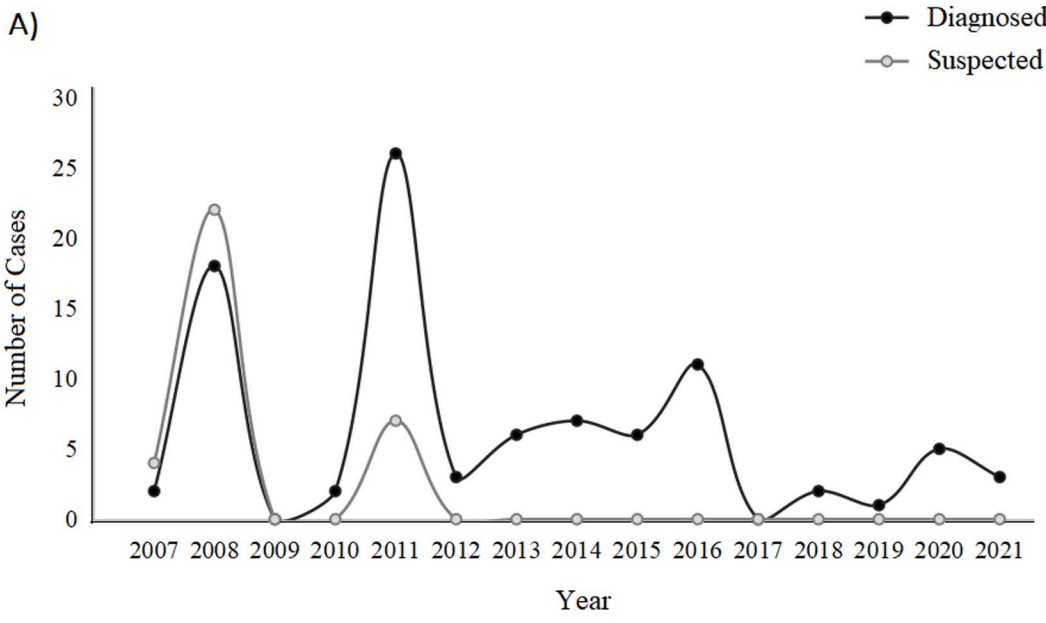

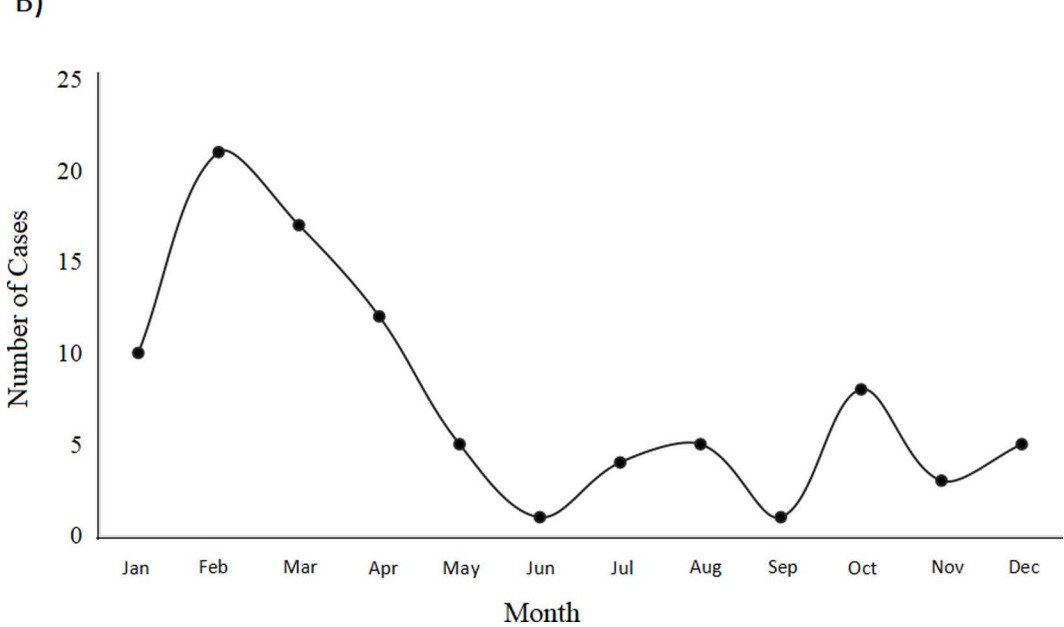

**Fig 1. (A)** Number of clinically diagnosed (n=92) and suspected (n=33) cases of human angiostrongyliasis in Dali from 2007 to 2021. **(B)** The total sum of diagnosed cases of human angiostrongyliasis (n=92) reported for each month in the Dali Prefecture from 2007 to 2021.

underwent investigation. The main clinical manifestations of the patients ranked by most to least common were headache, muscle pain, paresthesia, neck stiffness, fever, nausea, vomiting, and coughing (Fig 2). In addition, a small number of patients developed several different ocular symptoms such as photophobia (13 out of 72 cases), diplopia (18/72), and visual impairment (4/72).

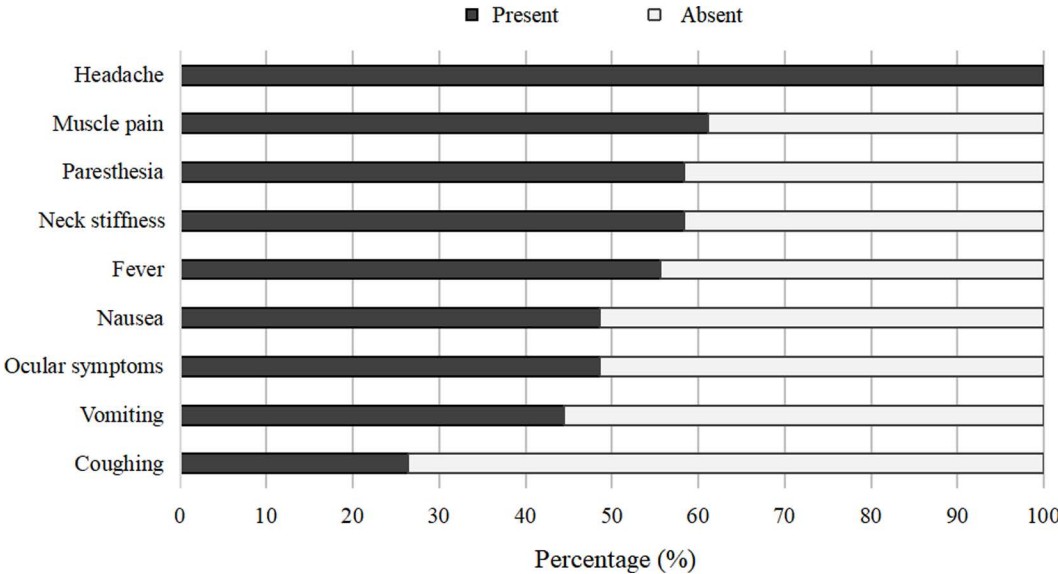

**Fig 2. The main clinical manifestations experienced by angiostrongyliasis patients (n = 72; mean age: 36 ± 13) in the Dali Prefecture from 2010 to 2021.**

## Laboratory test results

When diagnosing the patients, five different diagnostic tests were used to confirm the presence of *A. cantonensis* infection. However, each of the tests used showed different levels of accuracy in detecting abnormal results. A complete blood count test showed 87.5% (63/72) of patients had an abnormal number of eosinophils and 34% (21/72) of patients had an abnormal number of leukocytes (Table 1). Only 11.1% (8/72) of patients had both normal eosinophil and leukocyte levels. Similarly, when testing the cerebrospinal fluid (CSF) contents of the patients, we found that 97.5% (41/42) of patients had abnormal levels of eosinophils (Table 1). Additionally, we found that 90.5% (38/42) and 100% (39/39) of patients had an abnormal increase in protein content, and had abnormal qualitative protein levels, respectively (Table 1). However, the remaining test results were predominantly normal in the majority of patients. Only 35.7% (15/42) of patients exhibited abnormal chloride levels, followed by 16.6% (7/42) having abnormal CSF characteristics, 4.8% (2/42) having abnormal CSF pressure, and no patients had *Angiostrongylus* larvae present in their CSF (Table 1). A chest X-Ray examination of the patients showed the presence of abnormal shadows in their lungs, with only 2 of the 68 patients tested showing abnormal results (Table 1). Likewise, a head CT scan showed that 6 out of the 69 patients had abnormal results (Table 1), with the presence of elongated shadows and nodular enhancing lesions in the meninges of the brain and spinal cord. Lastly, 72 patients underwent ELISA testing, with only 34 of them showing positive results for *A. cantonensis* infection (Table 1).

## Prevalence of *A. cantonensis* infection in market sold *Pomacea* spp. snails

In addition to monitoring human cases of *A. cantonensis* infection, *Pomacea* spp. snails sold in Dali City wet market stalls were monitored for *A. cantonensis* infection from 2008–2021. In total, 49,970 commercially available *Pomacea* spp. were tested for *A. cantonensis* infection, with 373 (0.75%) snails being found infected. The highest number of *A. cantonensis* positive snails were found in 2010, with an infection prevalence of 1.66% (85/5124; Table 2; Fig 3a).

The monthly trend of *A. cantonensis* infected snails from 2009 to 2021 shows that infection is highest in February to May, with the highest incidence being in March (Fig 3b). However, it is still unclear what causes this seasonality increase in infection. The seasonal variation in *Pomacea* spp. infection prevalence could be driven by a combination of different

**Table 1. Number of Dali city angiostrongyliasis patients (mean age: 36±13) from 2010-2021 exhibiting normal or abnormal laboratory test results.**

| | | Normal results (n:) (Median, Range) | Abnormal results (n:) (Median, Range) | Total (n:) | Notes: |
|---|---|---|---|---|---|
| Blood count | Leukocytes (WBC) (×10⁹/L) | 51 (7.95, 4.64) | 21 (11.28, 36) | 72 | Leukocyte: (4–10 x10⁹/L) |
| | Eosinophils (EOS%) (%) | 9 (1.55, 3.30) | 63 (11.60, 72.31) | 72 | Percentage: (0.5~5%) |
| | Absolute Eosinophils (AEC) (×10⁹/L) | 9 (0.08, 0.33) | 63 (1.14, 14.09) | 72 | Eosinophil: (0-0.5 x10⁹/L) |
| Cerebrospinal fluid test | Pressure | 41 | 2 | 43 | – |
| | Characteristic | 36 | 7 | 43 | Abnormal: 6 cases of micro-mixing and 1 case of turbidity |
| | Protein content (mg/L) | 4 (218.75, 253) | 38 (621.00, 1315) | 42 | Protein content: 150–450mg/L |
| | Qualitative protein | 0 | 39 | 39 | Negative |
| | Chloride (mmol/L) | 27 (125.00, 11.55) | 15 (117.20, 8.90) | 42 | Chloride range: 120–132 mmol/L |
| | Eosinophils | 1 | 41 (38.00, 70) | 42 | Eosinophil range: <=0 |
| | *Angiostronglyus* larvae | 42 | 0 | 42 | – |
| X-Ray (Pulmonary shadow) | – | 66 | 2 | 68 | – |
| Head CT | Medulla oblongata | 63 | 6 | 69 | – |
| | Nodular enhancing lesions | 63 | 6 | 69 | – |
| ELISA | – | 38 | 34 | 72 | – |

Note: WBC: White Blood Cell, EOS%: Eosinophil Count Percentage, AEC: Absolute Eosinophil Count, CT: Computed Tomography, ELISA: Enzyme-Linked Immunosorbent Assay.

**Table 2. Prevalence of *Angiostrongylus cantonensis* infected *Pomacea* spp. snails sold at market in Dali City from 2008 to 2021.**

| Year | Number of batches | Snails tested | Infected | Positive (%) |
|---|---|---|---|---|
| 2008 | 23 | 2634 | 5 | 0.19 |
| 2009 | 40 | 2515 | 13 | 0.52 |
| 2010 | 58 | 5124 | 85 | 1.66 |
| 2011 | 60 | 5668 | 73 | 1.29 |
| 2012 | 36 | 2961 | 19 | 0.64 |
| 2013 | 50 | 3481 | 34 | 0.98 |
| 2014 | 57 | 4449 | 58 | 1.30 |
| 2015 | 52 | 3860 | 12 | 0.31 |
| 2016 | 44 | 3451 | 26 | 0.75 |
| 2017 | 54 | 4228 | 23 | 0.54 |
| 2018 | 46 | 2940 | 9 | 0.31 |
| 2019 | 44 | 2883 | 4 | 0.14 |
| 2020 | 44 | 3220 | 6 | 0.19 |
| 2021 | 46 | 2556 | 6 | 0.24 |

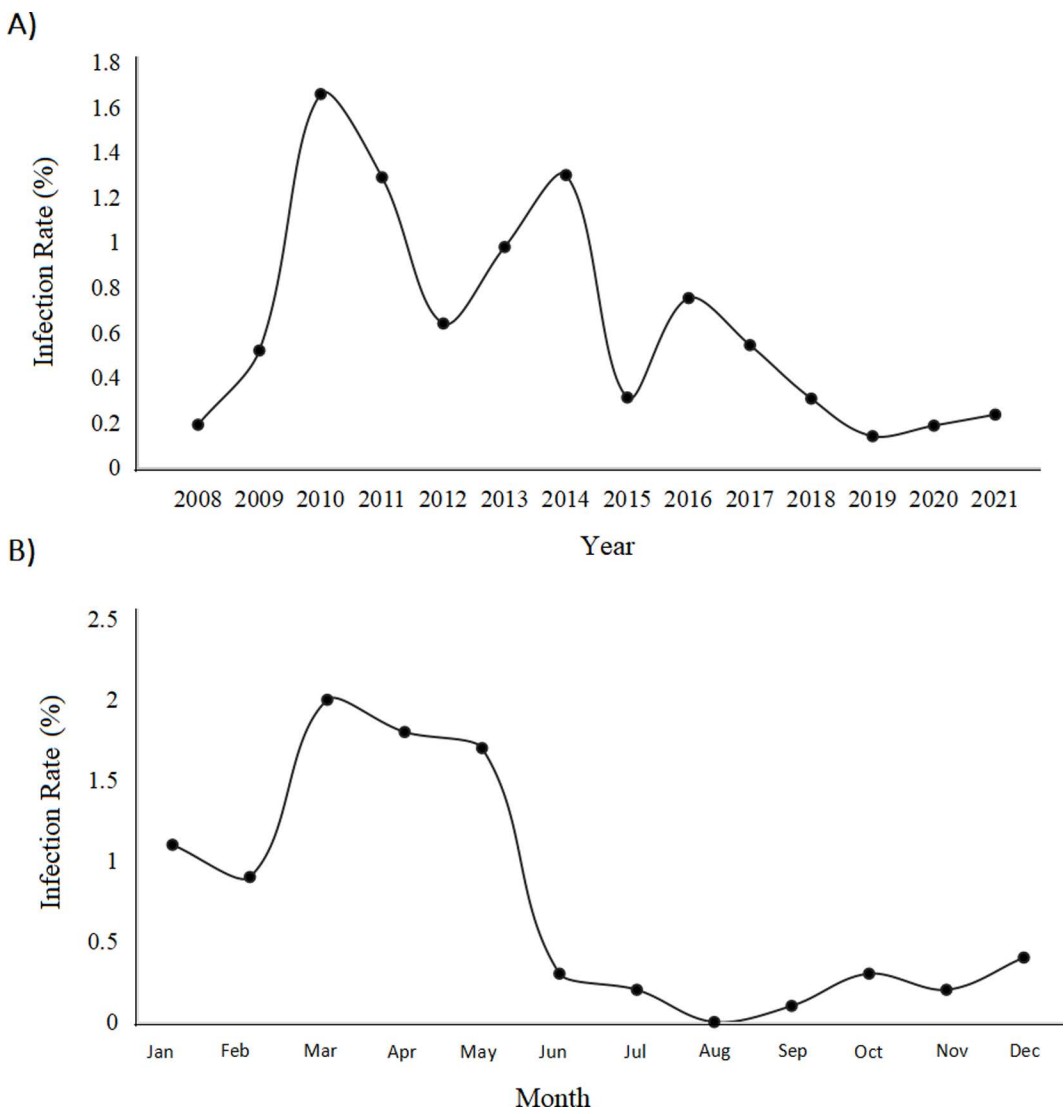

**Fig 3.  (A) Percentage of infected *Pomacea* spp. sold at market stalls in Dali City from 2008 to 2021; and (B) Monthly aggregated percentage of infected *Pomacea* spp. sold at market stalls in Dali City from 2008 to 2021.**

environmental factors, such as warmer temperatures and higher humidity starting in February, which promotes increased activity in both rodents and snails. During this period, favorable environmental conditions may lead to a rapid rise in rodent populations, potentially introducing more infective larvae into the environment and subsequently increasing infection rates in *Pomacea* spp. snails. However, it is uncertain what is causing this rapid increase in infection as the source of the snails purchased from the Dali market stalls is unknown and they could be a mixture of many different places (both farmed and wild). This highlights the need for stricter food safety practices to improve the standards of wet markets. Similarly, the observed seasonal increase in *Pomacea* spp. infection prevalence coincides with the higher incidence of reported angiostrongyliasis cases during spring (Figs 2b and 3b). Therefore, in order to prevent and control the recurrence of public health emergencies caused by *A. cantonensis* in the Dali area, it is necessary to strengthen the monitoring of snails from February to April. If the infection rate of snails is too high, relevant departments need to issue early warnings.

The overall prevalence of *A. cantonensis* infected snails showed a downward trend from 2010 to 2021, with a particularly noticeable decline in the past five years (Fig 3a). However, currently there is not enough supply of freshwater snails in the Dali area to meet the local demand. Most of the commercially available snails come from other places. In recent years, the source of the commercially available snails has been changed, which may be one of the reasons for the current infection. Although our data shows a reduction in *A. cantonensis* infection in both humans and snails in the Dali area in recent years, the risk of another angiostrongyliasis outbreak remains significant if the recently implemented safety measures are not maintained.

Of the 373 infected *Pomacea* spp. found, we extracted the DNA from the recovered *Angiostrongylus* larvae and amplified the COI gene fragment of 19 representatives from 2013 to 2021 (Fig 4a). We found the COI tree showed two groups, with the larvae from 2018 to 2021 being genetically different from the larvae collected from 2013 to 2016. However, when we amplified the ITS2 for 17 *A. cantonensis* representatives, we found all of the larvae showed little variation from other *A. cantonensis* isolates (Fig 4b). This is likely because the ITS2 gene fragment used in the analysis is more conserved than the COI region. The genetic difference observed in the COI between the *A. cantonensis* larvae collected before 2018 and those collected after 2018 is likely due to a change in the source of *Pomacea* spp. sold in the market (Fig 4a).

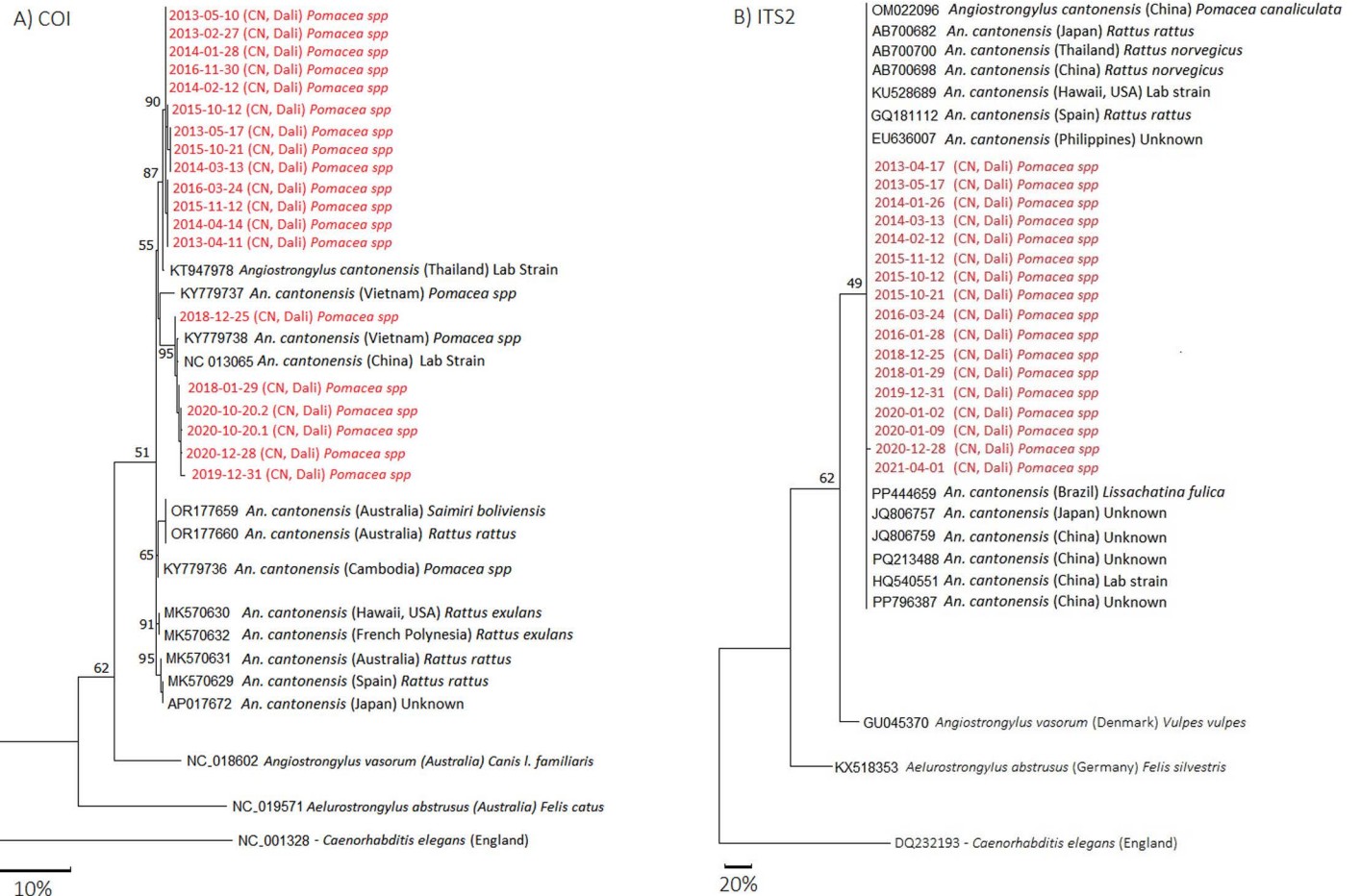

**Fig 4. Maximum likelihood tree of *A. cantonensis* larvae recovered from infected *Pomacea* spp. snails using the A) COI (460 bp) and B) ITS2 (304 bp) gene fragments.** Both trees were generated using MEGA 11 using a GTR+Γ model. Both trees included other lung worm species (*Angiostrongylus vasorum* and *Aelurostrongylus abstrusus*) and were rooted using *Caenorhabditis elegans*. Numbers on branches indicate the bootstrap percentages for 1000 replicates (bootstrap values under 50% were not shown). The scale bar represents sequence divergence.

### Prevention and control measures for wild *Pomacea* spp. snails

The government department has mobilized a physical prevention and control method by using local residents to manually collect *Pomacea* spp. snails and eggs. The peak period for egg laying by *Pomacea* spp. is in the spring and autumn seasons, as a response the artificial removal of *Pomacea* spp. egg masses is organized. The government departments of Eryuan County in 2009 and Dali City in 2021 have both implemented prevention and control measures that involved purchasing *Pomacea* spp. eggs. In the short term, the density of snails has decreased, achieving the goal of preventing further spread of *Pomacea* spp.

### Current understanding of angiostrongyliasis in Dali City

In response to the distribution of angiostrongyliasis in Guangzhou from 2010 to 2021, it was found that the people of Dali City have a weak awareness of the harm caused by *A. cantonensis*. Relevant departments regularly conduct awareness campaigns on angiostrongyliasis, educating the public about the role rats play in the transmission cycle of the disease, and the dangers of consuming terrestrial and freshwater gastropods. These efforts include broadcasting, online platforms, and distributing informational materials to provide knowledge on disease prevention and effective rat and snail control measures. For example, if a significant presence of rats or Pomacea spp. is observed in the area, it should be reported promptly. Moreover, local residents should be advised against consuming raw or undercooked gastropods and prawns, as this behavior can facilitate the transmission of *A. cantonensis*. However, changing this practice is challenging due to the regional culinary preferences, as seasoned raw snails and prawns are often preferred over thoroughly cooking them.

## Discussion

This study analyzed the number of human angiostrongyliasis cases reported in Dali hospitals from 2007 to 2021, surveying 72 of the 125 patients diagnosed with angiostrongyliasis. This observation of a large number of human angiostrongyliasis cases spanning over 14 years is rare, as *Angiostrongylus* infection often occurs in developing tropical countries and is frequently misdiagnosed. Human angiostrongyliasis is an emerging, neglected tropical disease, and this study offers insight into the clinical manifestations, diagnostic test performance, trends in infection prevalence and diagnostic challenges of angiostrongyliasis in Dali.

However, this study has several limitations. For example, this study did not have a cohort design as the patient data presented did not include a control group. Consequently, it was not possible to calculate the statistical significance of demographic factors (e.g., gender, age, employment) among the recorded patients. Similarly, this study only investigated the infection prevalence of *Pomacea* spp. snails purchased from two stalls at a single market in Dali, with no information on their origin. This provides only a limited epidemiological survey of the snails in the Dali region, and the lack of historical data on the sampled *Pomacea* spp. snails prevents the identification of potential environmental or human-mediated factors contributing to the variation in infection rate. Another limitation was the lack of recorded data on the onset of symptoms and the time of serum collection for each patient. This absence is significant, as the timing of serum collection relative to symptom onset is important for interpreting ELISA results, given that antibody levels can fluctuate over the course of infection. Moreover, this study did not use PCR detection methods to confirm the presence of *A. cantonensis* DNA in diagnosed patients. This is because *Angiostrongylus* infection in humans does not present with obvious signs such as eggs or larvae in the stool like other parasitological diseases, as humans are an accidental host. Instead, the adult worms often reside in the meninges, making it difficult to obtain *Angiostrongylus* DNA from patients. For example, molecular detection methods such as real-time PCR are able to detect *A. cantonensis* DNA in the cerebrospinal fluid and blood of infected patients [18].

### Human angiostrongyliasis cases and clinical diagnosis

Every year on the 8th of February, the Dali Prefecture residents celebrate the "snail festival" during the lunar new year [13], during this festival it is customary to eat raw snail meat, raw fish, and raw shrimps. Freshwater snails are captured

and released into wild water systems after the festival. However, due to multiple outbreaks of human angiostrongyliasis that occurred in the Dali Prefecture and other areas, it has had a significant change in the response of the local government to public health emergencies. Of the 125 angiostrongyliasis cases reported from 2007 to 2021, the largest outbreaks occurred in 2008 (diagnosed: 18, suspected: 22), 2011 (diagnosed: 26, suspected: 7) and 2016 (diagnosed: 11, suspected: 0), with only 11 cases being reported in the last five years (Fig 1a). The spring months (February: 21, March: 17, and April: 12) had the most reported cases (Fig 1b). Similarly, the highest prevalence of infected *Pomacea spp.* in Dali market stalls was in 2010 (1.66%), 2011 (1.29%) and 2014 (1.3%; Fig 3a), with the spring months (March: 2%, April: 1.9%, and May: 1.8%) having the highest number of infected snails (Fig 3b). This monthly trend of reported cases aligns with the annual snail festival and increased consumption of raw freshwater snails and prawns. However, the unknown origins of the market sold *Pomacea* spp. make it unclear why infections surged from 2010 to 2014, and every March to May.

When we looked at the locations for each of the cases reported, we found that Dali City and its urban areas exhibit the highest proportion of patients. This may be due to urban culinary preferences, or there are fewer rural monitoring hospitals that may result in missing possible cases. When we looked at the sex, age and profession for each of the patients, we did not observe any notable trends regarding the demographics of those infected. However, we did find a significant majority (89%) of patients had a history of consuming culinary dishes containing raw snail meat, or sauce fried snails. Therefore, given that nearly all angiostrongyliasis cases were linked to the local tradition of consuming raw or undercooked snail meat, it emphases the heightened risk of angiostrongyliasis in the Dali Prefecture and other regions. To mitigate further risk and prevent future angiostrongyliasis cases, it is crucial to educate and enhance public awareness of the disease, by modifying the dietary customs of local residents to avoid the health risks associated with consuming raw or undercooked snails. Furthermore, public education should go beyond dietary changes and should emphasize the importance of seeking medical attention if individuals experience severe headaches, a stiff neck and other symptoms after consuming raw or undercooked snails in order to achieve early diagnosis and treatment.

Of the 92 diagnosed angiostrongyliasis cases reported between 2007 and 2021, a subset of 72 cases were investigated to determine the most prevalent clinical manifestations of the disease. We found that severe headaches were experienced by all 72 patients, whereas coughing was only observed in 26% of patients (Fig 2). Other commonly reported symptoms, such as vomiting, ocular symptoms, nausea, fever, neck pain, paresthesia, and muscle pain, exhibited varying frequencies, with approximately 44% to 61% of patients experiencing them. Additionally, diagnostic accuracy varied across five lab tests for *A. cantonensis* infection. Of the five tests, the cerebrospinal fluid test demonstrated the highest accuracy, with 97.5% of tested patients exhibiting abnormal eosinophil levels, along with 90.5% and 100% of patients showing an abnormal increase in both protein content and qualitative protein levels, respectively. Similarly, the blood test revealed abnormal eosinophil levels in 87.5% of patients. However, chest X-rays, head CTs, and ELISA were inconsistent, with only 2.9% to 47.2% exhibiting abnormal results when using these tests.

The clinical symptoms commonly used to diagnose angiostrongyliasis lack specificity, as symptoms like headaches, muscle soreness, and neck stiffness are easily misdiagnosed, leading to potential under reporting of true cases [19]. Diagnosing *A. cantonensis* infection in humans is further complicated by the presence of atypical symptoms in some patients. Therefore, improving medical personnel training and expanding the access to advanced laboratory techniques are critical for improving diagnostic accuracy of human angiostrongyliasis cases. For example, real-time PCR should be adopted as the standard method for detecting *A. cantonensis* in suspected human cases, given its ability to identify infections in both cerebrospinal fluid and blood samples [18].

### *Pomacea* spp. and the future risk of angiostrongyliasis outbreaks in Dali

In 2003, *Pomacea* spp. were recognized as one of the 16 invasive snail species listed by China's Ministry of Ecology and Environment [20]. *Pomacea* spp. have become the main intermediate host for the transmission of angiostrongyliasis in China, with a review of published human angiostrongyliasis cases in the mainland of China suggesting that approximately

75% of the 334 cases reported before 2006 were directly related to the consumption of *Pomacea* spp. [21]. While other freshwater snails, terrestrial snails, semi-slugs and slug species can also transmit the disease, *Pomacea* spp. is of particular concern due to its popularity as an affordable and widely consumed food source. Furthermore, the Dali Prefecture presents various conditions that are conducive to the local spread and prevalence of angiostrongyliasis in the region. The ecological environment of the Dali Prefecture has numerous mountains and well-developed water systems, with a large distribution of wild *Pomacea* spp. around Erhai Lake [22]. These *Pomacea* spp. have been introduced from other regions, and when combined with the widespread distribution of rodents, there is a high likelihood of *A. cantonensis* invading new territories.

Since the first report of angiostrongyliasis in the Dali Prefecture at the end of 2007, we have consistently observed a high prevalence of *A. cantonensis* infected *Pomacea* spp. in Dali City market stalls for more than a decade. Apart from two outbreaks [13,23], there have been sporadic cases of angiostrongyliasis, with infected *Pomacea* spp. being concentrated in the first half of each year. Therefore, it is necessary to increase the monitoring of snails in the first six months of each year. At the same time, in response to two incidents of angiostrongyliasis outbreaks in Dali City, relevant departments should regularly conduct checks on restaurants selling snails in Dali City in order to reduce the risk of angiostrongyliasis and other food-borne public health emergencies. The precise source of these commercially available, *A. cantonensis* positive *Pomacea* spp. in Dali City remains unknown. However, when asked, vendors indicated *Pomacea* spp. before 2018 came from Ruili, and after 2018 came from Kunming, Jinghong, Sichuan, and other regions. Given the expanding presence and distribution of *Pomacea* spp. in the Dali Prefecture, coupled with ongoing occurrences of angiostrongyliasis cases, there is a notable risk of local angiostrongyliasis outbreaks in Dali. It is therefore important to investigate the distribution and infection patterns of *Pomacea* spp. near Erhai Lake and its surrounding areas.

In conclusion, the public health risk of angiostrongyliasis in Dali is closely linked to traditional dietary practices and unregulated wet markets in the region. The implementation of food safety measures and improved public education has significantly reduced the number of cases in recent years, effectively preventing large-scale outbreaks. However, the disease is still present and continues to pose a risk if these preventive measures are not maintained.

## Supporting information

**S1 Table. GenBank accession numbers for the COI and ITS2 phylogenetic trees.**
(DOCX)

## Acknowledgments

None.

## Author contributions

Tian-mei Li (TML) was responsible for the laboratory work, data processing, and the writing of the manuscript. Yun-hai Guo (YHG) and Shao-rong Chen (SRC) were responsible for the experimental design, and reviewing the manuscript. Yu-hua Liu (YHL) was responsible for manuscript revision. Wen Fang (WF), Shen-hua Zhao (SHZ), Ting Li (TL), and Ling Jiang (LJ) were involved in sample collection. Peter Andrus (PSA) assisted in the analysis, writing, reviewing, and correcting the grammar and spelling of the manuscript.

## Author contributions

**Conceptualization:** Tian-mei Li, Yunhai Guo, Shao-rong Chen.

**Data curation:** Wen Fang, Shen-hua Zhao, Ting Li, Ling Jiang.

**Formal analysis:** Tian-mei Li.

**Funding acquisition:** Tian-mei Li, Yunhai Guo.

**Investigation:** Wen Fang, Shen-hua Zhao, Ting Li, Ling Jiang.

**Supervision:** Yunhai Guo.

**Writing – original draft:** Tian-mei Li, Yu-hua Liu, Peter S Andrus, Yunhai Guo, Shao-rong Chen.

**Writing – review & editing:** Tian-mei Li, Yu-hua Liu, Peter S Andrus, Yunhai Guo, Shao-rong Chen.

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
