## [Decision Letter · Decision Letter 0]

24 Nov 2024

PNTD-D-24-01132Monitoring the trends of human angiostrongyliasis cases and Angiostrongylus cantonensis infected Pomacea snails in Dali, Yunnan, China from 2007 to 2021PLOS Neglected Tropical Diseases Dear Dr. Guo, Thank you for submitting your manuscript to PLOS Neglected Tropical Diseases. After careful consideration, we feel that it has merit but does not fully meet PLOS Neglected Tropical Diseases's publication criteria as it currently stands. Therefore, we invite you to submit a revised version of the manuscript that addresses the points raised during the review process. Please submit your revised manuscript within 60 days Jan 23 2025 11:59PM. If you will need more time than this to complete your revisions, please reply to this message or contact the journal office at plosntds@plos.org. Please include the following items when submitting your revised manuscript: * A rebuttal letter that responds to each point raised by the editor and reviewer(s). You should upload this letter as a separate file labeled 'Response to Reviewers '. This file does not need to include responses to any formatting updates and technical items listed in the 'Journal Requirements' section below. * A marked-up copy of your manuscript that highlights changes made to the original version. You should upload this as a separate file labeled 'Revised Manuscript with Track Changes '. * An unmarked version of your revised paper without tracked changes. You should upload this as a separate file labeled 'Manuscript '. If you would like to make changes to your financial disclosure, competing interests statement, or data availability statement, please make these updates within the submission form at the time of resubmission. Guidelines for resubmitting your figure files are available below the reviewer comments at the end of this letter. We look forward to receiving your revised manuscript. Kind regards,Siddhartha Mahanty, M.B.B.S., M.P.HAcademic EditorPLOS Neglected Tropical Diseases Jong-Yil ChaiSection EditorPLOS Neglected Tropical Diseases 

Shaden Kamhawi

co-Editor-in-Chief

Paul Brindley

co-Editor-in-Chief

**Additional Editor Comments :** All the reviewers acknowledged the strengths of your study, particularly the large number of cases, comprehensive clinical and laboratory data and the linkage of human infections wth prevalence of A. cantonensis in Pomacea spp. However, some issues with serological and molecular methodology and epidemiological design have been identified by the reviewers that should be addressed for the masncrupt to be considered for publication. Please address all the reviewers' comments paying particular attention the issues raised by reviewers 1 and 3. **Journal Requirements:** 

At this stage, the following Authors/Authors require contributions: Tian-mei Li, Yu-hua Liu, Wen Fang, Shen-hua Zhao, Ting Li, Ling Jiang, Yun-hai Guo, and Shao-rong Chen. Please ensure that the full contributions of each author are acknowledged in the "Add/Edit/Remove Authors" section of our submission form.

2) The uploaded Manuscript file contains highlights in the Ethics statement section. Please provide a clean version of your manuscript from your latest manuscript version.

3) Please provide an Author Summary. This should appear in your manuscript between the Abstract (if applicable) and the Introduction, and should be 150u2013200 words long. The aim should be to make your findings accessible to a wide audience that includes both scientists and non-scientists. Sample summaries can be found on our website under Submission Guidelines:

4) Please upload a copy of Figures 1b, 2b, 3b, and 4b which you refer to in your text on pages 7, 9, 10, and 11. Or, if the figure is no longer to be included as part of the submission please remove all reference to it within the text.

5) Please ensure that all Figure files have corresponding citations and legends within the manuscript. Currently, Figures 5, and 6 in your submission file inventory do not have in-text citations. If the figure is no longer to be included as part of the submission, please remove it from the file inventory.

6) Tables should not be uploaded as individual files. Please remove these files and include the Tables in your manuscript file as editable, cell-based objects. For more information about how to format tables, see our guidelines:

https://journals.plos.org/plosntds/s/tables 

7) We have noticed that you have uploaded Supporting Information files, but you have not included a list of legends. Please add a full list of legends for your Supporting Information files after the references list.

8) Thank you for stating that : "All data is given within the manuscript itself." Please confirm at this time whether or not your submission contains all raw data required to replicate the results of your study. Authors must share the “minimal data set” for their submission. PLOS defines the minimal data set to consist of the data required to replicate all study findings reported in the article, as well as related metadata and methods (https://journals.plos.org/plosone/s/data-availability#loc-minimal-data-set-definition).

9) Please amend your detailed Financial Disclosure statement. This is published with the article. It must therefore be completed in full sentences and contain the exact wording you wish to be published.

2) State what role the funders took in the study. If the funders had no role in your study, please state: "The funders had no role in study design, data collection and analysis, decision to publish, or preparation of the manuscript.".

**Reviewers' Comments:**Reviewer's Responses to Questions

**Key Review Criteria Required for Acceptance?**

**Methods**

-Are the objectives of the study clearly articulated with a clear testable hypothesis stated?

-Is the study design appropriate to address the stated objectives?

-Is the population clearly described and appropriate for the hypothesis being tested?

-Is the sample size sufficient to ensure adequate power to address the hypothesis being tested?

-Were correct statistical analysis used to support conclusions?

-Are there concerns about ethical or regulatory requirements being met?

Reviewer #1: -Are the objectives of the study clearly articulated with a clear testable hypothesis stated? Yes

-Is the study design appropriate to address the stated objectives? Yes

-Is the population clearly described and appropriate for the hypothesis being tested? Yes

-Is the sample size sufficient to ensure adequate power to address the hypothesis being tested? Yes

-Were correct statistical analysis used to support conclusions? No

-Are there concerns about ethical or regulatory requirements being met? No

Far more detail of the ELISA employed is required. What antigen was used? Was this an IgG or IgM ELISA? Was it a commercial kit or in-house developed assay? What controls were employed? What OD cut-off was used? Is there a reference to any validation data for this assay?

How many weeks or months after the onset of symptoms was serum drawn for ELISA testing? Please add the mean and range (in days) for all samples tested by this method.

What positive and negative controls were employed in the CO1 and ITS2 PCRs?

Lines 121-122: Please briefly summarise what diagnostic methods are recommended in the PRC Health Industry Standards (WS321-2010), so that readers understand exactly what diagnostic methods were employed in the cases reported.

Line 148: Was Sanger sequencing used?

Lines 154-156: How many bootstraps were used in the neighbour joining tree.

Lines 162-171: What were the odds ratios of male vs female for confirmed angiostrongyliasis cases? What about the chi-squared or Fisher exact test analysis of age brackets (0-5 years old, 5-10 years old, 10-15 years old, >15 years old)? Odds ratio for specific employment types? Odds ratio of eating specific high-risk foods? Please perform more detailed statistical analyses to determine if any of the reported factors were statistically significant.

Reviewer #2: The authors present a very interesting article detailing the cases of angiostrongyliasis diagnosed in their area from 2008-2021 along with testing of snails for this pathogen. Many of the cases are from an outbreak of the parasite that occurred in 2007-2008.

They reviewed all human cases of angiostrongyliasis in their Prefecture, although they do not include their selection criteria for cases or how cases were identified. The also screened snails in wet markets with monthly testing.

Reviewer #3: Yes, good study design and a very comprehensive study.

**Results**

-Does the analysis presented match the analysis plan?

-Are the results clearly and completely presented?

-Are the figures (Tables, Images) of sufficient quality for clarity?

Reviewer #1: -Does the analysis presented match the analysis plan? Yes

-Are the results clearly and completely presented? No

-Are the figures (Tables, Images) of sufficient quality for clarity? No

I cannot find figure legends anywhere in the manuscript – please include these. The figure legend for figure 4 should include what method and software was used to prepare the tree, and how many bootstrap replicates were performed. I would like to review the details of the legends once they have been included in the manuscript.

Please change the x axis (months) labels in figures 1b and 3b from numbers to abbreviated month names (e.g. Jan, Feb, Mar, etc.)

The data provided in Figure 4 (the phylogenetic tree) is insufficient. Some data from the supplementary data table should be included here after the Genbank accession numbers. Please include: city of isolation, date of isolation, source (Pomacea spp., human, etc.). Note that the outlier is Caenorhabditis elegans. A second outlier which is phylogenetically closer to A. cantonensis should be included also, I suggest Aelurostrongylus abstrusus or Angiostrongylus vasorum.

Please also add more sequences from global A. cantonensis isolates to figure 4 to ensure that the Yunnan genotypes can be visualised in the context of global genotypes. I suggest adding at least a sequence of A. cantonensis from each continent (North America, South America, Europe, Africa, Australia-Pacific, somewhere in Asia other than China).

Table 1: Please add the number (n) and age (mean average and range) of patients from which samples were included in this analysis in the table legend.

Please add the median value and range of indices reported in table 1

Please add units for all indices reported in table 1. Use S.I. units: https://www.nist.gov/pml/owm/metric-si/si-units

For precision of analysis, peripheral eosinophil count in Table 1 should be reported as whole numbers and not as a percentage of the white cell count. The generally accepted normal range for eosinophils is ≤0.5 x10^9 cells/L. Please reanalyse your data using an internationally recognised normal eosinophil range in cells/L

Lines 188-189: 87.5% of patients had peripheral eosinophilia and only 34% of patients exhibited leukocytosis. Please explicitly state how many patients had normal FBC results (no leukocytosis or eosinophilia).

Lines 210-212. Please expand on the seasonality observed in snail infections. What is the climate like in these months in the area the snails come from? Winter? Summer? Wet? Dry? Describe this and please consider how this might affect rat and snail behaviour and populations, then provide some suggestions as to why these months saw such a spike in mollusk infections. Could seasonal variation in food availability (and consequent abundance) and degree of activity of reservoir host rats influence the seasonality of intermediate host of snail infections from source regions? Could seasonal variability in the abundance and activity of intermediate host snails influence the seasonality of intermediate host of snail infections from source regions?

Reviewer #2: The authors present clinical and demographic data for the 125 cases of Angiostrongylus infection they identified, along with the percent of snails tested that were positive for the parasite.

Reviewer #3: Yes, very well-presented and well-written

**Conclusions**

-Are the conclusions supported by the data presented?

-Are the limitations of analysis clearly described?

-Do the authors discuss how these data can be helpful to advance our understanding of the topic under study?

-Is public health relevance addressed?

Reviewer #1: -Are the conclusions supported by the data presented? Yes

-Are the limitations of analysis clearly described? No

-Do the authors discuss how these data can be helpful to advance our understanding of the topic under study? Somewhat - more detail could be included

-Is public health relevance addressed? Somewhat - more detail could be included

This manuscript's greatest value is the large number of angiostrongyliasis cases identified, as such a large cohort is difficult to obtain, so the clinical indices data is very useful. This should be highlighted more.

As you do not have molecular results (CSF PCR) confirming that all of your cases are truly angiostrongyliasis, how can you be sure that the diagnoses are correct? Please discuss this limitation thoroughly

What were the limitations of the study performed? Please briefly discuss these.

Why were only Pomacea spp. snails sampled? Are these the only snails or mollusks sold for consumption in Dali? Please explain the choice of Pomacea spp. only in the manuscript.

Sampling only two food stalls is not a valid epidemiological survey of snails. While the data is worth including, the epidemiological limitations of this sampling approach must be discussed.

Line 316-320: The statement made in this section is untrue. The CDC in the United States runs a real time PCR for A. cantonensis on CSF and a positive result is a definitive indicator if A. cantonensis central nervous system infection. This must be acknowledged and discussed. Refer to: Qvarnstrom Y, Xayavong M, da Silva AC, Park SY, Whelen AC, Calimlim PS, Sciulli RH, Honda SA, Higa K, Kitsutani P, Chea N. Real-time polymerase chain reaction detection of Angiostrongylus cantonensis DNA in cerebrospinal fluid from patients with eosinophilic meningitis. The American journal of tropical medicine and hygiene. 2016 Jan 1;94(1):176.

Reviewer #2: The authors noted a clear seasonality for when cases seem to occur each year, with most cases in February - April. They also provide interesting data regarding diagnostic testing for angiostrongylus, which shows the widely varying utility of varying diagnostic tests.

Reviewer #3: Yes, very comprehensive and an important study

**Editorial and Data Presentation Modifications?**

Reviewer #1: Line 251: Why only freshwater snails? Terrestrial snails, slugs and semi-slugs are all intermediate hosts of A. cantonensis, why were these not included. What about public health control of rats?

Line 256: I have not previously encountered the term “Guangzhou roundworm” referring to A. cantonensis. Please add the common name (rat lungworm) in brackets after this name.

Line 351: It is not really imperative that this investigation be performed, please soften this wording.

Please add further discussion on the role of rats in the A. cantonensis life cycle to the introduction.

Please write “Pomacea spp.”, not just “Pomacea” throughout

Note that while in entomology and malacology often the first two letters of a genus are included in abbreviation, in parasitology this is not done. Please change all “An. cantonensis” to “A. cantonensis” throughout the manuscript.

Some of the epidemiological discussion in the manuscript is unnecessarily verbose and could be rewritten in a more succinct manner.

A short concluding paragraph summarising the most important findings of this study is necessary.

The title could be shortened to something like, “Monitoring the trends of Angiostrongylus cantonensis in humans and Pomacea snails, Dali, Yunnan, China, 2007-2021

Reviewer #2: (No Response)

Reviewer #3: (No Response)

**Summary and General Comments**

Reviewer #1: This report summarises data on human cases of angiostrongyliasis from 2007 to 2021 in Yunnan, China. The high number of cases described gives this report merit, as larger scale clinical data such as this is difficult to obtain. The survey of gastropods from food sellers is adjunct, and also has merit. My major criticisms are that additional details of the methods employed are required, as well as ranges for the reported clinical data. I have further, more minor comments, detailed below.

The authors may wish to follow up on IgG serology for these patients in another study. A major gap in our understanding of A. cantonesis diagnostics is how many patients mount an IgG antibody response, and how long tis response lasts for. A longitudinal study of A. cantonesis IgG titres in a large cohort such as this would be very helpful for clinical diagnostics and surveillance purposes.

Reviewer #2: Overall, this was an interesting and well written paper that presents useful information regarding the presentation of a generally neglected parasitic infection, Angiostrongylus.

Reviewer #3: A very substantial study and well-presented. I have only minor mostly grammatical suggestions outlined below.

Over a very well-written and significant paper. This is one of few papers that documents snail infection rates and corroborates it with human infection temporally (over an extensive time period). I have only minor suggestions outlined below. A question I have, and likely not possible with this large dataset, is if there are subsets of human samples available to run PCR? Also, I noticed the annealing temperature used for PCR of snails is very low. Is there a reason for this? I am assuming cross-reactivity would be picked up by sequencing, so I am wondering if the annealing temperature was evaluated, and how was 48C selected. The authors might want to include a statement in the methods. Also, I was a bit surprised of the relatively low infection rates in snails.

L30 analyze

L56 gastropod

L57 A. cantonensis not An. cantonensis?

L60 Might want to include Hawaii

L98 outline

L101 Is the ethics statement properly placed within the manuscript?

L145 48C seems really low. All products were sequenced?

L161 cases

L193 content

L206 In total, 49,970 …..

L239 collect Pomacea snails and eggs.

L281 different from

L316-318 Why not use PCR of CSF or blood as a confirmatory test?

PLOS authors have the option to publish the peer review history of their article (what does this mean? ). If published, this will include your full peer review and any attached files.

**Do you want your identity to be public for this peer review?** For information about this choice, including consent withdrawal, please see our Privacy Policy .

Reviewer #1: **Yes: ** Richard Bradbury

Reviewer #2: No

Reviewer #3: No

**Figure resubmission:** While revising your submission, please upload your figure files to the Preflight Analysis and Conversion Engine (PACE) digital diagnostic tool, https://pacev2.apexcovantage.com/. PACE helps ensure that figures meet PLOS requirements. To use PACE, you must first register as a user. Registration is free. Then, login and navigate to the UPLOAD tab, where you will find detailed instructions on how to use the tool. If you encounter any issues or have any questions when using PACE, please email PLOS at figures@plos.org. Please note that Supporting Information files do not need this step. If there are other versions of figure files still present in your submission file inventory at resubmission, please replace them with the PACE-processed versions.**Reproducibility:** To enhance the reproducibility of your results, we recommend that authors of applicable studies deposit laboratory protocols in protocols.io, where a protocol can be assigned its own identifier (DOI) such that it can be cited independently in the future. Additionally, PLOS ONE offers an option to publish peer-reviewed clinical study protocols. Read more information on sharing protocols at https://plos.org/protocols?utm_medium=editorial-email&utm_source=authorletters&utm_campaign=protocols

---

## [Decision Letter · Decision Letter 1]

23 Mar 2025

PNTD-D-24-01132R1Monitoring the Trends of Angiostrongylus cantonensis Infection in Humans and Pomacea spp. Snails in Dali, Yunnan, China, 2007-2021PLOS Neglected Tropical Diseases Dear Dr. Guo, Thank you for submitting your manuscript to PLOS Neglected Tropical Diseases. After careful consideration, we feel that it has merit but does not fully meet PLOS Neglected Tropical Diseases's publication criteria as it currently stands. Therefore, we invite you to submit a revised version of the manuscript that addresses the points raised during the review process.

Please submit your revised manuscript within 30 days Apr 22 2025 11:59PM. If you will need more time than this to complete your revisions, please reply to this message or contact the journal office at plosntds@plos.org. Please include the following items when submitting your revised manuscript:

* A rebuttal letter that responds to each point raised by the editor and reviewer(s). You should upload this letter as a separate file labeled 'Response to Reviewers '. This file does not need to include responses to any formatting updates and technical items listed in the 'Journal Requirements' section below.

* A marked-up copy of your manuscript that highlights changes made to the original version. You should upload this as a separate file labeled 'Revised Manuscript with Track Changes '.

* An unmarked version of your revised paper without tracked changes. You should upload this as a separate file labeled 'Manuscript '.

We look forward to receiving your revised manuscript.

Kind regards,

Siddhartha Mahanty, M.B.B.S., M.P.H

Academic Editor

Jong-Yil Chai

Section Editor

Shaden Kamhawi

co-Editor-in-Chief

Paul Brindley

co-Editor-in-Chief

 **Additional Editor Comments :**

The reviewers noted that the manuscript has been considerably modified and improved to address their main concerns but there are some less critical changes recommended by them that, if addressed, would raise the quality of the paper considerably. Please address the issues mentioned in their summary comments and in an attachment from Reviewer 3. Please respond to these issues for the manuscript to be acceptable for publication in the Journal.

**Comments to the Authors:**

**Please note that the reviews are uploaded as attachments.**

**Reviewers' comments:**

Reviewer's Responses to Questions

**Key Review Criteria Required for Acceptance?**

**Methods**

-Are the objectives of the study clearly articulated with a clear testable hypothesis stated?

-Is the study design appropriate to address the stated objectives?

-Is the population clearly described and appropriate for the hypothesis being tested?

-Is the sample size sufficient to ensure adequate power to address the hypothesis being tested?

-Were correct statistical analysis used to support conclusions?

-Are there concerns about ethical or regulatory requirements being met?

Reviewer #1: (No Response)

Reviewer #3: Yes, but no comment on the statistical methods.

**Results**

-Does the analysis presented match the analysis plan?

-Are the results clearly and completely presented?

-Are the figures (Tables, Images) of sufficient quality for clarity?

Reviewer #1: (No Response)

Reviewer #3: It's much improved

**Conclusions**

-Are the conclusions supported by the data presented?

-Are the limitations of analysis clearly described?

-Do the authors discuss how these data can be helpful to advance our understanding of the topic under study?

-Is public health relevance addressed?

Reviewer #1: (No Response)

Reviewer #3: Yes, again much improved

**Editorial and Data Presentation Modifications?**

Reviewer #1: (No Response)

Reviewer #3: Minor revision, I don't need to see again if my suggested changes are made.

**Summary and General Comments**

Reviewer #1: The authors have satisfactorily addressed all of my review comments.

Note on taxonomic writing: Throughout the revised version, "spp." is italicised (i.e. "Pomacea spp."). Only full Latin and Greek genus/species names need to be italicised. As this is an abbreviation of an English word ('species"), it should not be italicised.

Reviewer #3: I believe the authors adequately addressed the reviewer's concerns. Is there no way to retrieve the dates of blood draw? This would significantly strengthen this ms.

PLOS authors have the option to publish the peer review history of their article (what does this mean? ). If published, this will include your full peer review and any attached files.

**Do you want your identity to be public for this peer review?** For information about this choice, including consent withdrawal, please see our Privacy Policy .

Reviewer #1: **Yes: ** Richard Stewart Bradbury

Reviewer #3: No

**Figure resubmission:**
---

## [Editor Report · Decision Letter 2]

16 Apr 2025

Dear Professor Guo,

We are pleased to inform you that your manuscript 'Monitoring the Trends of Angiostrongylus cantonensis Infection in Humans and Pomacea spp. Snails in Dali, Yunnan, China, 2007-2021' has been provisionally accepted for publication in PLOS Neglected Tropical Diseases.

Best regards,

Siddhartha Mahanty, M.B.B.S., M.P.H

Academic Editor

Jong-Yil Chai

Section Editor

Shaden Kamhawi

co-Editor-in-Chief

Paul Brindley

co-Editor-in-Chief

---

## [Editor Report · Acceptance letter]

Dear Professor Guo,

We are delighted to inform you that your manuscript, "Monitoring the Trends of Angiostrongylus cantonensis Infection in Humans and Pomacea spp. Snails in Dali, Yunnan, China, 2007-2021," has been formally accepted for publication in PLOS Neglected Tropical Diseases.

Best regards,

Shaden Kamhawi

co-Editor-in-Chief

Paul Brindley

co-Editor-in-Chief
